# Environmentally Friendly Improvement of Plasmonic Nanostructure Functionality towards Magnetic Resonance Applications

**DOI:** 10.3390/nano13040764

**Published:** 2023-02-17

**Authors:** Miroslava Flimelová, Yury V. Ryabchikov, Jan Behrends, Nadezhda M. Bulgakova

**Affiliations:** 1HiLASE Centre, Institute of Physics of the Czech Academy of Sciences, Za Radnicí 828, 25241 Dolní Břežany, Czech Republic; 2Berlin Joint EPR Lab., Fachbereich Physik, Freie Universität Berlin, Arnimallee 14, 14195 Berlin, Germany

**Keywords:** ultrashort pulse laser ablation in liquid, gold–silicon nanoparticles, nanohybrids, hybrid nanomaterials, paramagnetic defects, plasmonic nanomaterials, magnetic resonance

## Abstract

Plasmonic nanostructures have attracted a broad research interest due to their application perspectives in various fields such as biosensing, catalysis, photovoltaics, and biomedicine. Their synthesis by pulsed laser ablation in pure water enables eliminating various side effects originating from chemical contamination. Another advantage of pulsed laser ablation in liquids (PLAL) is the possibility to controllably produce plasmonic nanoparticles (NPs) in combination with other plasmonic or magnetic materials, thus enhancing their functionality. However, the PLAL technique is still challenging in respect of merging metallic and semiconductor specific features in nanosized objects that could significantly broaden application areas of plasmonic nanostructures. In this work, we performed synthesis of hybrid AuSi NPs with novel modalities by ultrashort laser ablation of bulk gold in water containing silicon NPs. The Au/Si atomic ratio in the nanohybrids was finely varied from 0.5 to 3.5 when changing the initial Si NPs concentration in water from 70 µg/mL to 10 µg/mL, respectively, without requiring any complex chemical procedures. It has been found that the laser-fluence-insensitive silicon content depends on the mass of nanohybrids. A high concentration of paramagnetic defects (2.2·× 10^18^ spin/g) in polycrystalline plasmonic NPs has been achieved. Our findings can open further prospects for plasmonic nanostructures as contrast agents in optical and magnetic resonance imaging techniques, biosensing, and cancer theranostics.

## 1. Introduction

Plasmonic nanomaterials are attracting great research interest in the multifaceted upcoming field of nanomedicine due to the possibility of multifold enhancement of optical signals [1,2,3,4,5,6,7,8]. However, they lose implementation niches where semiconductor nanomaterials play a significant role covering many applications such as singlet oxygen generation, cancer theranostics, optical nanothermometry, and multimodal linear/nonlinear optical bioimaging [9,10,11,12,13,14,15,16,17,18,19]. Moreover, silicon-based nanostructures can also serve as efficient drug nanocarriers providing a simultaneous ultrasound-activated antimicrobial effect [20,21].

To improve nanomaterials’ functionality, many efforts are made by chemical means for merging plasmonic, magnetic, and semiconductor materials in one multifunctional nanoplatform. As a result, bimetallic nanoparticles were synthesized, which demonstrate promising features for the field of nanomedicine, in particular, as theranostic and imaging nanoagents [22,23]. Moreover, metal-semiconductor multielement nanomaterials have also been fabricated in the form of either alloy or core–shell nanostructures with a tuneable aspect ratio between their cores and shells. Multielement tailoring of nanomaterials allows considerable widening of their applications in many fields such as sensor, drug delivery, imaging, and therapeutic applications [24,25,26,27]. To further advance a great potential of hybrid nanomaterials, new routes of their facile synthesis are in great demand.

By now most processes of nanomaterial preparation are based on complex synthesis procedures including potentially toxic precursors. This obviates their applications, which require ultra-high purity materials, pushing the development of new simple techniques of contamination-free synthesis of hybrid nanostructures. To overcome the above-mentioned issues and ensure a high chemical purity of hybrids, preparation of nanomaterials is performed in biocompatible gaseous or liquid environments by using “green” laser-based methods. In particular, pulsed laser ablation in liquids (PLAL) has proven to be a very effective tool of the production of single-element plasmonic [28,29,30] and semiconductor silicon-based [31,32,33,34,35] nanoparticles with tuneable properties. Moreover, the efficient production of bimetallic nanostructures was achieved by laser-induced merging of plasmonic and magnetically active elements [36,37,38,39,40]. Nevertheless, the design of metal-semiconductor hybrid nanostructures using laser-based methods is an insufficiently explored niche in the modern nanotechnology, considering its great application prospects. We would like to emphasize the recently demonstrated potential of such laser-synthesized nanohybrids [41,42,43,44,45,46,47].

Here, we report on a facile approach for improving the functionality of plasmonic nanoparticles avoiding any complex reactions and chemical surfactants. A successful combination of metallic and semiconductor elements in one nanoparticle was achieved by ultrashort PLAL of bulk gold in water containing silicon nanoparticles. The produced nanohybrids possess polycrystalline gold structure with silicon impurities and exhibit simultaneously plasmonic and paramagnetic modalities. A procedure for fine tuning both the chemical composition and the size distribution of nanohybrids by variation of the concentration of silicon nanoparticles in a reactive medium was developed. A correlation of the chemical composition with the mean size of the gold–silicon nanohybrids was established. Our findings open new avenues for plasmonic-based nanostructures for optical and magnetic resonance bioimaging, biosensing, and cancer theranostics.

## 2. Materials and Methods

To improve the functionality of plasmonic nanoparticles, we employed the PLAL technique by irradiation of a gold target (purity of 99.999%) in deionized water-based environment with 30 nm spherical Si NPs whose concentration was varied in the range of 0–70 mg/L (Figure 1) [33]. Bare Au NPs were obtained by PLAL without Si NPs while the presence of Si NPs in water led to the formation of hybrid nanostructures. A femtosecond laser (PHAROS from Light Conversion, 250 fs pulse duration, 1030 nm wavelength, 1 kHz repetition rate) with the fluence controlled by a half-wave plate (100 and 150 µJ/pulse) was used. The position of the laser beam focused on the sample surface into a 50 µm diameter spot was controlled by a galvanoscanner with 2 m/s scanning speed within the irradiation area of 10 × 10 mm^2^. The gold samples placed under 10 mm water level were irradiated for 20 min. Large NPs, which were found to be formed in a small amount, were removed by a size separation step using the centrifugation at 10,000× *g* for 10 min [48].

To estimate the size distribution and the chemical composition of the Au-based NPs, a high-resolution transmission electron microscope (HR-TEM, Jeol-2011, Jeol Ltd, Tokyo, Japan) with energy-dispersive X-ray spectrometer (EDX) was used. Prior to the investigation, a drop of a freshly prepared NP suspension was deposited on a carbon-coated copper grid and dried under ambient conditions. The chemical composition of the gold–silicon nanohybrids (AuSi NHs) was studied by inspecting a single isolated nanoparticle of a given size. The nanoparticle size distribution was calculated by means of the ImageJ software (version 1.53t, National Institutes of Health and the Laboratory for Optical and Computational Instrumentation (LOCI, University of Wisconsin)) using approximately 1000 particles for each sample.

To study the structural properties of the produced nanohybrids, X-ray diffraction (XRD, Rigaku SmartLab, Rigaku Corporation, Tokyo, Japan) and X-ray Photoelectron Spectroscopy (XPS, ESCALAB 210 system, Thermo Scientific, Waltham, Massachusetts, USA) methods were used. The XRD data were collected in the angle range between 20° and 90° using a Cu Kα 0.1506 nm wavelength energy source. The grazing angle of the incident radiation was 0.5° that was larger than the critical angle for the total external reflection of the silicon substrate 0.23°. The angular range of the detector was calibrated measuring lanthanum hexaboride as an XRD standard. We studied XPS spectra of gold-based nanoparticles drop casted onto a Cu substrate (~1 mm^2^) in an ultrahigh vacuum (~10^−10^ Torr) by irradiation with a Mg K_α_ line (1253.6 eV) as a source of the photoelectrons.

To discover the functionality improvement of the plasmonic nanoparticles, we studied their optical and paramagnetic properties by means of UV-Vis (Shimadzu 2700, Shimadzu Corporation, Kyoto, Japan) and Electron Paramagnetic Resonance (EPR, laboratory-built spectrometer based on components from Bruker Biospin, Ettlingen, Germany) spectroscopies, respectively. The first one allowed us to check plasmonic properties of the nanohybrids whereas the second one provides information on the nature and concentration of paramagnetic defects. The EPR experiments were performed on a laboratory-built spectrometer described in [49] (the main parameters are the following: modulation amplitude of 0.5 mT, modulation frequency of 100 kHz, microwave power attenuation of 20 dB, temperature of 100 K).

## 3. Results

To analyze the size distribution, the shape, and the crystalline structure of laser-synthesized NPs without and with silicon impurities, the dried nanoparticle samples were characterized by means of transmission electron microscopy. We found a strong impact of Si NPs suspended in the ablation medium on the size of nearly round-shaped hybrid nanostructures ranging from several to one hundred nanometres (Figure 2).

Moreover, much more homogeneous size dispersion of hybrid AuSi NPs compared to bare ones was observed (Figure 2a,b). Nevertheless, their diffraction patterns revealed several diffraction rings, pointing to identical crystalline structure of NPs regardless the presence of silicon species (Figure 2c,d). This indicates an impurity-independent structure of the plasmonic-based nanomaterials with environment-sensitive size distributions (see Discussion section for details).

The size distributions of nanoparticles synthesized in pure water and in water with suspended Si NPs (30 mg/L) at two different laser pulse energies are presented in Figure 3. Bare Au NPs exhibit bimodal size distributions with the mean sizes in the first and second peaks, respectively, of ~15 nm and ~26 nm at 100 µJ/pulse, and ~18 nm and ~34 nm at 150 µJ/pulse. Their full width at half maximum (FWHM) showed broadening > 30% when the irradiation energy was increased by 50%: ~10.5 nm and ~21 nm at 100 µJ/pulse; ~14 nm and ~31 nm at 150 µJ/pulse. The presence of two peaks in the size distributions may indicate that different mechanisms of nanoparticle formation upon laser ablation in the case of bare Au NPs (see [50] and Section 4). It can be speculated that, by adding Si NPs in the reactive solution, some mechanisms are suppressed. Indeed, hybrid AuSi NPs exhibit only one peak in the size distribution, which is narrower as compared to bare Au NPs. In particular, the FWHM decreased from ~10.5 nm for bare Au NPs to ~8 nm for NHs at 100 µJ/pulse and from ~14 nm to ~9 nm, respectively, at 150 µJ/pulse. Interestingly, the FWHM is found to be almost fluence-independent for hybrid nanostructures. Moreover, the same mean size (~12 nm) of AuSi NHs was observed for nanomaterials synthesized using two different laser fluences. These findings indicate a strong sensitivity of the photochemical reactions occurring during formation of Au-based NPs to their molecular environment.

To analyse the chemical composition of the AuSi nanostructures of different sizes, HR-TEM-EDX studies using individual NPs of a given diameter were carried out. Large NPs (>25 nm) revealed a major gold contribution (~98% of mass) independent on the laser fluence (Figure 4a,b). However, decreasing the diameter below 25 nm leads to a significant change in the gold content from 98% of mass for 25 nm to 63% of mass for 5 nm of hybrid NPs. It is worth noting that the size-dependent chemical composition was not affected by the change in the laser fluence (Figure 4). These results can help to estimate and finely tune the chemical composition of multicomponent Au-based NPs based on their size variation.

To explore the influence of reactive medium, the dependences of both the chemical composition and the mean size of the hybrid NPs on the concentration of Si NP in water were analysed (Figure 5). It has been found that they were identical at different laser fluences. Moreover, with increasing the concentration of the Si NPs, the ratio of numbers of gold and silicon atoms (Au/Si ratio) reduces exponentially from 3.5 (10 μg/mL) to 0.5 (70 μg/mL) (Figure 5a). At the same time, the mean size of laser-synthesized NPs decreases from 19 nm at 10 μg/mL to 8 nm at 70 μg/mL (Figure 5b). The equal amount of gold and silicon atoms was achieved in 11 nm AuSi NHs formed at 35 μg/mL Si NP concentration in water. These observations allow us choosing an appropriate concentration of Si NPs in order to synthesize AuSi NHs of a given size or with a certain ratio between atoms.

To study the structure and the phase composition of AuSi NHs, XRD, and XPS analyses were employed. Several clear maxima were found in the XRD patterns at different 2Θ values reflecting the crystalline structure of formed nanohybrids (Figure 6a). The most pronounced peaks were observed at 38.4°, 44.8°, 64.8°, 77.5°, and 82.3° and also some responses are seen at 36.3°, 41.7°, and 47.9° pointing to a polycrystalline nature of the nanostructures. The presence of different phases of nanostructured elements in the hybrid nanoparticles was analysed using XPS. Analysing the Au 4f spectrum, it was concluded that there is a possibility of several charge states of gold in the hybrid NPs. In order to identify them, the spectrum was deconvoluted by assuming the presence of two doublets with different full widths at half maximum (1.21 eV and 3.00 eV). Afterwards, each doublet was fitted by two Gaussian curves resulting in four signals with the following binding energy values: 80.1 eV and 84.3 eV (the first doublet) as well as 83.1 eV and 86.8 eV (the second doublet) (Figure 6b). The ratio between their integral intensities allows estimating the contribution of each charge state: 1.0:0.8:0.7:0.6.

After deconvolution of the Si 2p spectrum, the presence of a main peak located at a binding energy around 103.3 eV was observed together with a weak feature (of six times lower intensity) at 101.3 eV. These first results of the structural investigation of AuSi NHs can help in understanding the element redistribution and suggesting possible structures and content of laser-synthesized gold–silicon nanohybrids.

To confirm the combination of gold and silicon modalities in one NP, the nanohybrids were investigated by means of UV-Vis and EPR spectroscopies (Figure 7). The first method revealed plasmonic properties whereas the second one was employed to study paramagnetic defects in the formed nanostructures. Here, an obvious plasmonic response from both hybrid and bare NPs was found, which arises from nanostructured gold. In the absorbance spectrum of AuSi NHs, the plasmonic maximum was 18 nm blue-shifted as compared to bare Au NPs.

Additionally, AuSi NHs possess a more pronounced absorbance in the blue part of the spectral range, most probably due to the presence of silicon species. At the same time, EPR spectra of the gold-based NPs (measured at 100 K to improve the signal-to-noise ratio) reveal a strong difference between structures of nanomaterials produced with and without Si NPs in water environment (Figure 7b). A remarkable EPR response from nanohybrids containing silicon impurities indicates the presence of paramagnetic defects with unpaired electrons. Neither the intensity nor the position of the EPR signals changed upon varying the laser fluence used for synthesis of the nanohybrids. However, no EPR signal (only a noise level even at low temperature) within the same magnetic-field range was detected from the bare Au NPs prepared at different laser fluences, pointing out the absence of such type of defects. These results unequivocally justify the possibility of easy surfactant-free combination of several modalities in one nanoparticle, which is a crucial point for extending their applications.

## 4. Discussion

The improvement of the nanostructure functionality for performing a wide range of tasks in the fields of biomedical applications, nuclear medicine, molecule sensing, catalysis, and photovoltaics is a challenging issue nowadays. For this purpose, it is important to develop a simple, contamination-free, time- and cost-effective method of NP synthesis suitable for various materials. As demonstrated here, laser ablation of gold in water containing silicon nanoparticles represents the fast controllable way to combine, in a nanohybrid form, metal and semiconductor materials having different modalities significant for biomedical applications. The method offers the possibility of varying both the chemical composition and the size distribution of the nanostructures, thus adjusting their properties. One of the key points is the use of surfactant-free “green” synthesis conditions that is important for biomedical, sensing, and catalytic tasks. A puzzling question is a narrow single-peak size distribution with the fluence-independent FWHM for the hybrid nanostructures, contrary to the bare Au NPs produced under the same irradiation conditions (Figure 3). To address this question, we consider below the established picture of the PLAL with involving some concepts on the states and interactions of atoms and nanoparticles both in plasma within the bubble and dissolved in water.

Laser ablation of solids in a liquid environment is a complicated phenomenon, which involves many processes, starting from energy absorption by a target material, its heating, melting, and ablation that results in the formation of a cavitation bubble, where the ablation products expand to. At ultrashort pulse laser irradiation, the ablation products represent a mixture of nanodroplets and atomic/molecular fraction, which subsequently evolve with formation of the final population of nanoparticles suspended in liquid [29,51]. The size distribution, stoichiometry, and chemical composition of the NPs significantly depend on the ablated material, medium, and irradiation parameters. In particular, laser irradiation of a gold target in deionized water results in formation of bare Au NPs with a fluence-dependent size distribution when the increase in laser fluence leads to appearance of larger Au NPs [50]. As a whole, broad size distributions, where the desired small nanoparticles coexist with larger ones, are commonly observed in laser ablation in liquid environments regardless of a pulse duration [50,52].

As was already mentioned, the bimodal size distribution of Au NPs (Figure 3) can point out a coexistence of different mechanisms of nanoparticle formation. Recently by large-scale atomistic simulations of laser ablation of metals in liquid, three mechanisms of NP generation were identified [50], which include (1) nucleation and growth of small NPs via condensation inside the cavitation bubble, (2) spinodal decomposition of ablation products, leading to the formation of a large population of NPs through inter-particle collisions and coalescence, (3) decomposition of a transient metal layer formed at the interface between the ablation plume and liquid resulting in injection of large NPs into water beyond the cavitation bubble boundary. As a whole, the cavitation bubble serves as a reaction chamber for nanoparticle nucleation, growth, coalescence and solidification [51]. In this “chamber”, large Au NPs may mostly originate from the transient layer decomposition and/or coalescence of smaller NPs. It is then necessary to analyse why these mechanisms can be suppressed for the case of ablation in water containing Si NPs.

The presence of Si NPs in the ablation environment may considerably influence nanoparticle growth dynamics due to the following processes. First, the Si NPs partially absorb laser radiation. As a result, the laser beam is attenuated along its pass to the target that leads to a smaller laser fluence on the Au surface, thus reducing the resulting NP size distribution towards a single peak as reported in [50]. The second reason can be related to nanoparticle composition. It can be hypothesized that laser-induced decomposition of Si NPs in water with creation of atomic and small cluster fragments provokes the interaction between gold and silicon atoms with formation of Au–Si bonds. We suppose that this leads to reducing the size of hybrid nanoparticles (Figure 3) similar to the case described in the Ref. [53]. The presence of organic molecules such as dextran or cyclodextrin during laser ablation process of a gold target significantly changes the nanoparticles size. In particular, it was found that a 100-times increase in the concentration of cyclodextrin leads up to 3–5 times decrease in the size of Au NPs [53]. Similarly, the amount of small silicon fragments formed by laser induced decomposition of Si NPs may also influence the size of the formed nanohybrids. Indeed, a higher concentration of silicon particles suspended in water suppresses stronger the agglomeration efficiency between gold clusters, provoking a remarkable narrowing of the resulting size distribution of hybrid nanostructures (Figure 5b) with, at the same time, enhanced silicon content (Figure 5a).

From the above discussion, one can make a conclusion about an important role of the photochemical and kinetic processes occurring between gold and silicon during the formation of AuSi NHs and that the formation is not finished in the “bubble reaction chamber” and can proceed further in the liquid phase where the ablation products are suspended. Below we shortly discuss the main kinetic aspects of cluster/nanoparticle formation.

In the stage of bubble formation till its collapse [54], the main kinetics of NP formation proceeds in a vapor phase with partially ionized ablation plasma. This stage of the “reaction chamber” is quite long, up to several dozens of microseconds. As a result, for achieving a high throughput of NP with high-repetition-rate lasers, a high-speed scanning technique is needed to avoid overlapping of laser pulses with the bubbles created by previous pulses. In our case of a relatively low-repetition-rate laser, the bubbles are collapsing withing the time interval between pulses and beam scanning is used for avoiding creation of deep craters upon the synthesis process. The partially ionized ablation plume consists of gold vapor phase, gold nanoclusters, and droplets originated from the different regions of the developing bubble [51], which are evolving during expansion, being confined in the water surrounding. Water vapor and, in our case, atoms, ions, and clusters of silicon are also present in the “bubble reaction chamber” that adds complexity to the kinetics of NPs/NHs growth. In laser plasmas, metal clusters and small NPs are usually negatively charged due to their electron affinity and high mobility of electrons [55]. As a result, they electrostatically attract positive plasma ions (in our case, both Au^+^ and Si^+^), thus facilitating the growth of nanoparticles. We assume that the first hybrid AuSi nanoparticles are formed within the bubbles. However, their further growth dynamics may proceed in the liquid phase.

As already mentioned, when absorbing laser radiation, the NPs suspended in water experience efficient fragmentation [56] that is confirmed by their absence in the NP size distributions (Figure 3). According to atomistic simulations, the light absorbing nanoparticles are fragmented via the phase explosion mechanism with ejection atoms and atomic clusters and small droplets into surrounding liquid [57]. The vaporized Si atoms dissolve in water in a neutral state via the hydrolysis reaction with forming silicic acid [58]. The charge of the gold nanoparticles suspended in water is known to be negative due to surface hydroxylation followed by a loss of H^+^ [59]. This negative charge together with the electrostatic double layer effect inherent for colloidal nanoparticles [60] makes them stable against aggregation. We also suggest that silicic acid, which should be abundant in solution due to efficient laser-induced disintegration of Si NPs into atomic phase, may serve as a ligand, similarly to other acidic ligands [61]. However, formation mechanisms of PLAL-produced hybrid NPs are still not completely understood as they can involve nonequilibrium element-dependent chemistry and thermodynamic critical phenomena [62] that calls for further studies.

The crystalline structure of the laser-synthesized nanohybrids reveals several gold atomic planes as follows from the analysis of diffraction patterns (Figure 2c,d) and from the XRD data (Figure 6a). These results are summarized in Table 1 being in the good agreement with the literature data. Here, we can assume that the laser-synthesized NHs contain several gold nanodomains with different crystallographic orientation. We also hypothesize on the formation of gold–silicide nanoformulations of different stoichiometry and crystallinity considering additional small XRD responses in the 30°–60° 2Θ range (Figure 6a) [63]. Further evidence comes from the scattering from amorphous Au–Si obtained upon annealing of Au/Si thin multilayers found at the same 2Θ positions [64,65,66].

In spite of clear signals of crystalline gold, no traces of crystalline silicon in these data suggest that Si is present in the amorphous phase, thus leading to the absence of any Raman responses. This is in contradiction to the data published in the Ref. [44] where the optical signal can originate from bare Si NPs due to a poor solvent purification. A possible structure of AuSi NHs is suggested by estimating the sizes of different nanodomains (*d*) by using a Scherer formula:(1)$$d=\frac{K\lambda }{\beta cos\Theta }$$
where *K* is a correction factor depending on different parameters (crystallographic direction, shape, and distribution of the clusters) that is approximately 0.9 in our case [67], *λ* is the wavelength of the X-ray radiation, 2Θ is the Bragg angle, *β* is the full width at half maximum of the peak in radians [67]. It leads to the following values of nanodomains: ~5.7 nm for Au(111), ~3.2 nm for Au_5_Si_2_, and ~5.0 nm for AuSi, which are lower than sizes obtained from the TEM study. Hence, one can conclude that AuSi NHs consist of several interconnected crystalline nanodomains of different phases. However, understanding of the formation mechanisms of gold–silicide phases and their stoichiometry requires a separate study in order to establish conditions of their formation.

**Table 1 nanomaterials-13-00764-t001:** Interatomic distances of gold-based nanoparticles measured in the present work as compared to the literature data [68] (on the left), crystalline planes (in the middle), experimental values of the XRD peaks of AuSi NHs as compared to the literature data for gold NPs [69] (on the right).

Interplanar Distance (Å)			Planes	XRD Peaks (°)	
Au NPs	AuSi NHs	Literature		Experiments	Literature
2.38	2.39	2.35	(111)	38.4	38.2
2.12	2.08	2.04	(200)	44.8	44.4
1.47	1.48	1.44	(220)	64.8	64.8
1.26	1.26	1.23	(311)	77.5	77.6
1.19	1.19	1.18	(222)	82.3	82.0
0.95	-	0.94	(331)	-	-

The phase composition of the nanohybrids was analysed using XPS (Figure 6b,c). No Si^0^ state in the hybrid nanoparticles is expected due to the absence of any traces at a binding energy of around 99.0 eV [69,70]. However, the existence of some oxidation states of silicon can be expected in AuSi NHs. Indeed, signals at 101.3 eV and 103.3 eV in the Si 2p spectrum (Figure 6c) can be attributed to Si^+2^ and Si^+4^ oxidation states, respectively [67,68,69,70,71]. Contrary to silicon, the broad doublet at 83.4 eV may point towards the presence of Au^0^ metallic nanodomains [69,71,72]. However, the 0.6 eV shift compared to the literature value may indicate a charge redistribution between gold and silicon atoms because of different electronegativity values (2.5 and 1.9, respectively) [72]. We believe that it originates from Au–Si bonds formed by a laser-induced interaction between gold and silicon [69,71,72,73], confirming the formation of gold silicide nanoformulations. Moreover, the appearance of reduced oxidation states of gold atoms (Au^−2^ and Au^−4^) in the nanohybrids are downshifted by ~2 and ~4 eV with respect to that of metallic Au. Moreover, a direct connection seems to exist between the relative magnitude of the XPS signal intensity of the Si 2p peak arising from Si^+2^ and that of the Au 4f peak originating from the reduced Au. These facts suggest that the Si 2p and Au 4f core levels are naturally reflecting the oxidation states of the atomic elements composing the produced AuSi NHs. These results allow evidencing that a strong charge-transfer process may occur between Si and Au atoms during AuSi NH formation.

The successful combination of gold and silicon elements in one nanoparticle also follows from the absorbance and EPR studies. Both types of Au-based NPs have clear plasmonic maxima (Figure 7a) while the observed 18 nm blue shift of the plasmonic maximum of AuSi NHs can be associated with lower mean size of NPs. Their higher absorbance in the shorter wavelength range can indirectly indicate the presence of nanostructured silicon having a similar absorbance value [33]. Another direct evidence of the presence of silicon content in the synthesized nanomaterial follows from the EDX data giving us its quantitative value for nanoparticles of different sizes (Figure 4b). Si contribution is also directly confirmed by EPR measurements revealing a strong response in the 331–338 mT magnetic field range. A computer simulation of the experimental spectra gives us the g-factor value of 2.0055 ± 0.0005, which obviously originates from dangling bonds in disordered silicon with 2.2 × 10^18^ spin/g concentration [33]. No responses from the bare Au NPs even at low temperature justify its silicon origin. Along with TEM, XRD, and XPS data, it is another indication of the amorphous structure of silicon nanodomains in NHs.

Based on the results summarized above, we propose the following structure of gold–silicon nanohybrids prepared by laser ablation. The synthesized composite gold–silicon nanoparticles represent combinations of different nanodomains such as metallic gold of various crystallinity responsible for plasmonic properties, gold silicide of different stoichiometry as well as SiO and SiO_2_ reflecting paramagnetic features of the nanohybrids. Our results indicate that the developed method extends the functionality of plasmonic metal nanostructures by laser-induced combination with semiconductor materials, thus allowing their non-invasive tracking by magnetic resonance imaging techniques. As an outlook, we assume that it can provide further significant extensions of the performance of hybrid metallic-semiconductor nanoparticles by the development of multifunctional nanotools due to their combination with other elements. For instance, laser-induced incorporation of different types of radionuclides (RNs) can add new modalities to laser-synthesized nanostructures such as diagnostic and/or therapeutic one by choosing an appropriate type of elements (Figure 8). It can considerably facilitate the formation of radionuclide nanocarriers for nuclear nanomedicine that are currently performed by complexed chemical methods [74,75] where RNs can play a dual role (both diagnostics and therapeutic), locally detecting or treating a malignant tumour. Together with surfactant-free synthesis allowing chemical composition variation, such ultrapure multicomponent nanoagents can play a very important role in many biomedical applications such as bioimaging, biosensing, hyperthermia, nuclear medicine, photodynamic, and radio-therapy of cancer (Figure 8) [76,77].

## 5. Conclusions

In summary, a cost- and time-effective method of extending the functionality of plasmonic nanoparticles is demonstrated. A successful combination of both plasmonic and paramagnetic modalities is achieved in hybrid nanomaterials manufactured by a “green” laser-based technique. Their polycrystalline gold structure containing gold–silicide nanoformulations and silicon impurities is observed. A protocol of a tiny adjustment of both chemical composition and size distribution of nanohybrids requiring neither complex reactions nor toxic precursors has been developed. Accordingly, the chemical composition of the synthesized AuSi NHs is adjustable in the range of 0.5–3.5 Au/Si atomic ratio. The size-dependent increase in silicon content from 2% to 35% of mass of nanohybrids is found to be fluence-insensitive. A mean diameter of narrowly-dispersed AuSi nanostructures is easily varied in the range of 8–18 nm. Further prospects of the developed technique towards manufacturing of multifunctional nanoagents combining multiple modalities in one nanoparticle suitable for solving several biomedical tasks simultaneously are suggested.

## Figures and Tables

**Figure 1 nanomaterials-13-00764-f001:**
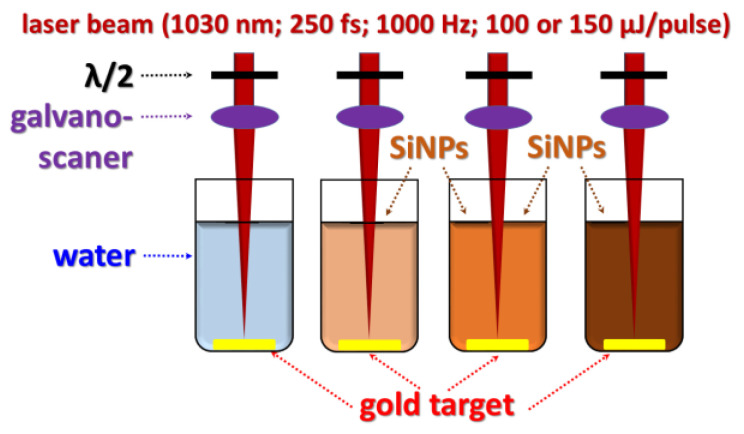
A scheme of PLAL synthesis of gold-based nanoparticles.

**Figure 2 nanomaterials-13-00764-f002:**
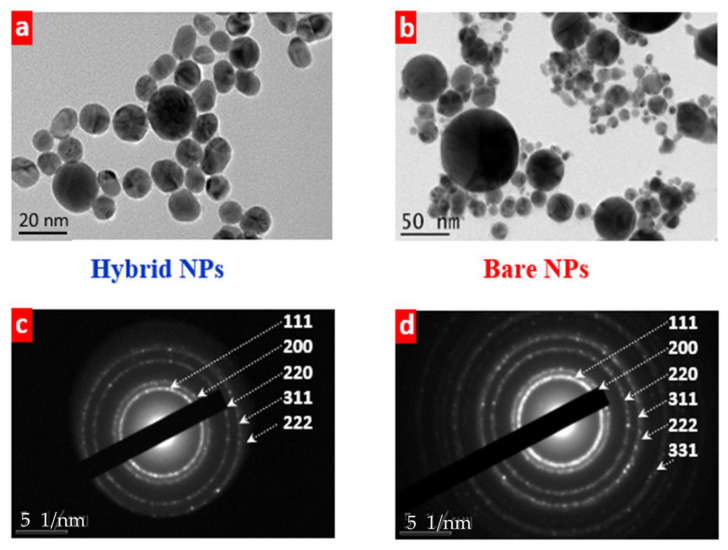
TEM images (**a**,**b**) and diffraction patterns (**c**,**d**) of hybrid AuSi (**a**,**c**) and bare Au (**b**,**d**) NPs. Laser fluence 100 µJ/pulse, initial concentration of Si NPs (**a**,**c**)—30 mg/L.

**Figure 3 nanomaterials-13-00764-f003:**
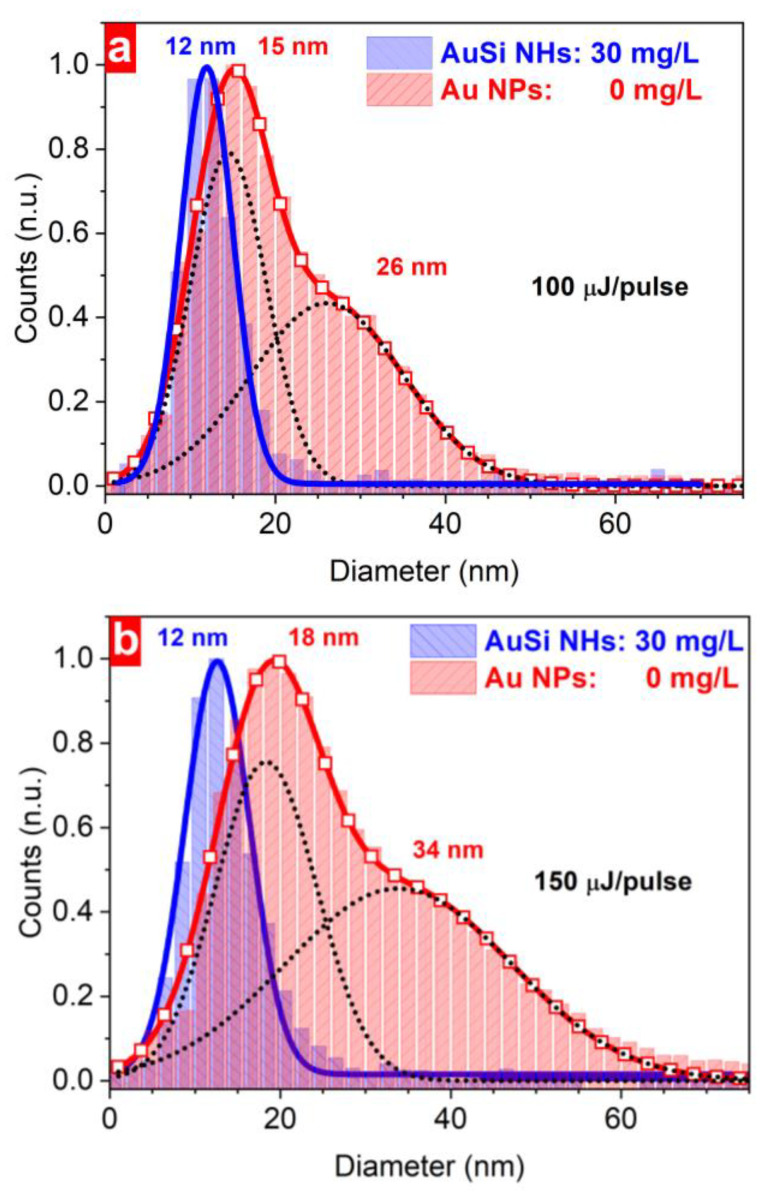
Size distributions of AuSi hybrid and bare Au NPs formed at laser fluences of 100 µJ/pulse (**a**) and 150 µJ/pulse (**b**). Hybrid NPs were obtained in 30 mg/L solution of Si NPs in water. Dotted lines correspond to fitting the Au NPs distributions by two Gaussian curves.

**Figure 4 nanomaterials-13-00764-f004:**
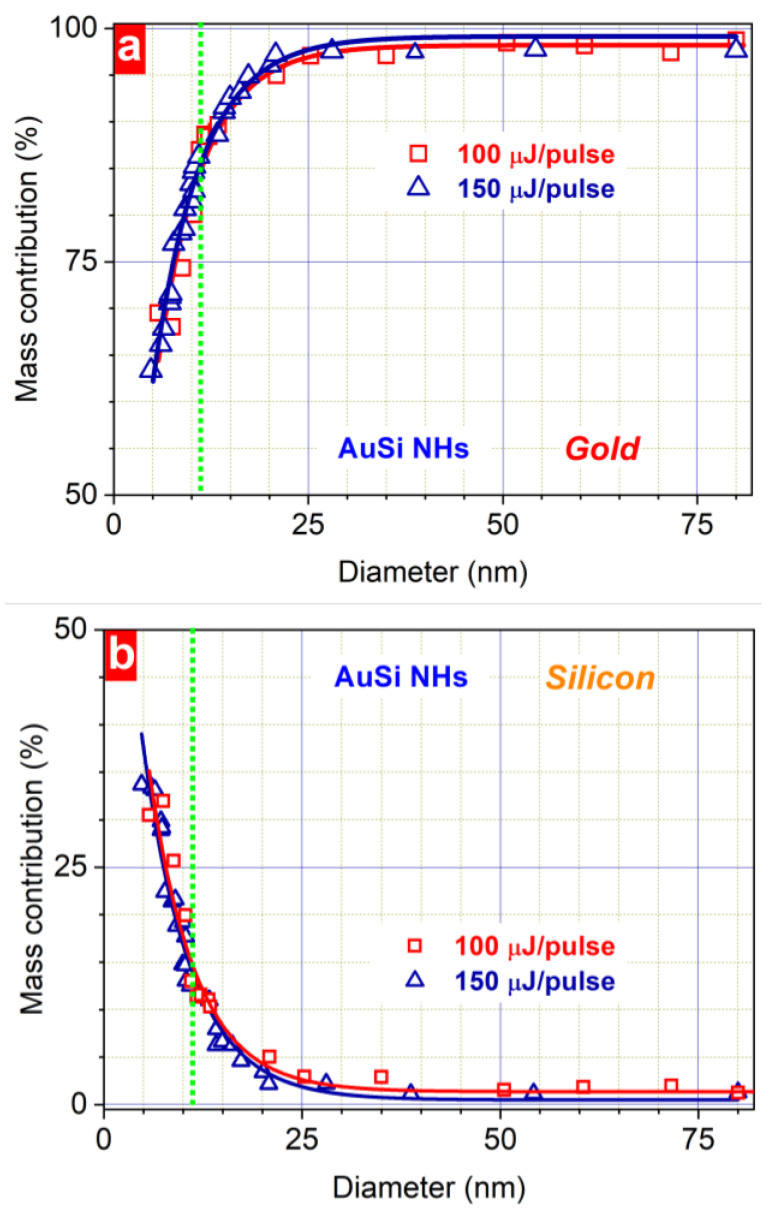
Size-dependent gold (**a**) and silicon (**b**) mass content in AuSi NHs prepared at 100 µJ/pulse and 150 µJ/pulse laser fluence. The green dotted lines correspond to the size of AuSi NHs containing equal number of Au and Si atoms.

**Figure 5 nanomaterials-13-00764-f005:**
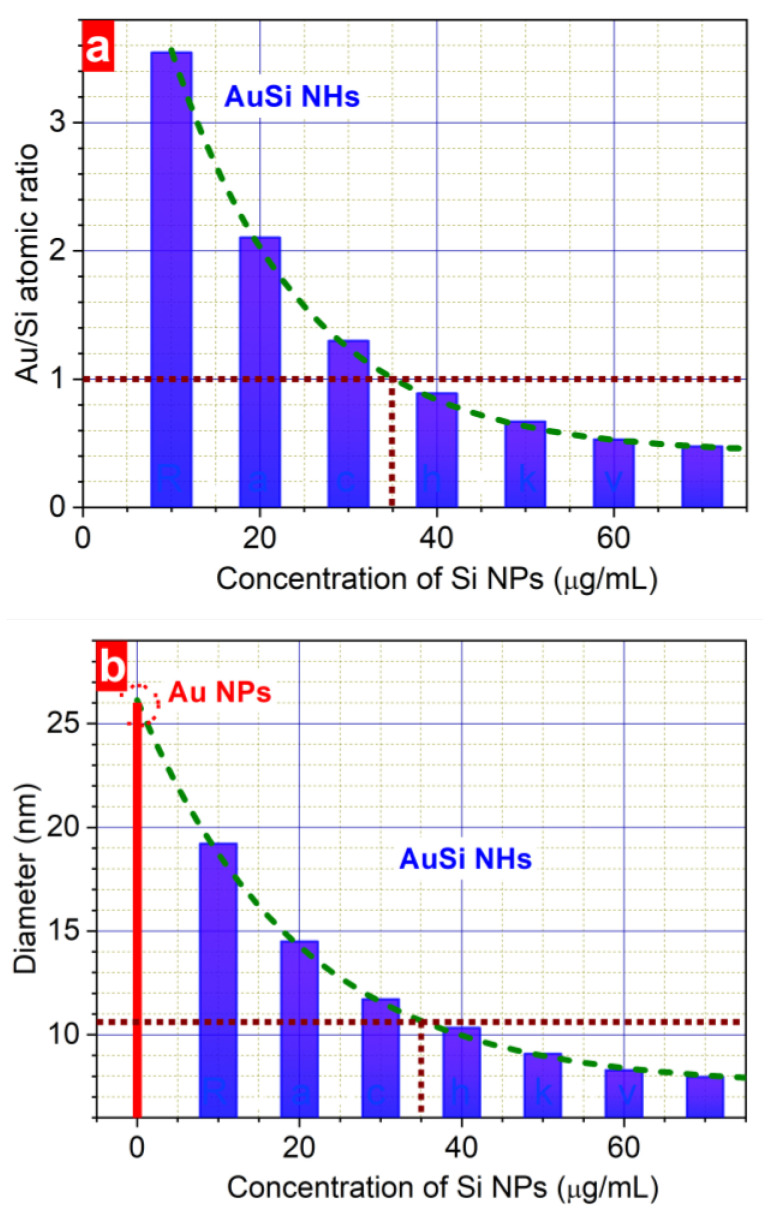
Dependence of the chemical composition (**a**) and mean size (**b**) of synthesized nanoparticles on the concentration of Si NPs in water. Laser fluence is 100 μJ/pulse. The vertical brown dotted lines correspond to the concentration of Si NPs used for preparation of AuSi NHs with the average size ~11 nm containing equal number of Au and Si atoms.

**Figure 6 nanomaterials-13-00764-f006:**
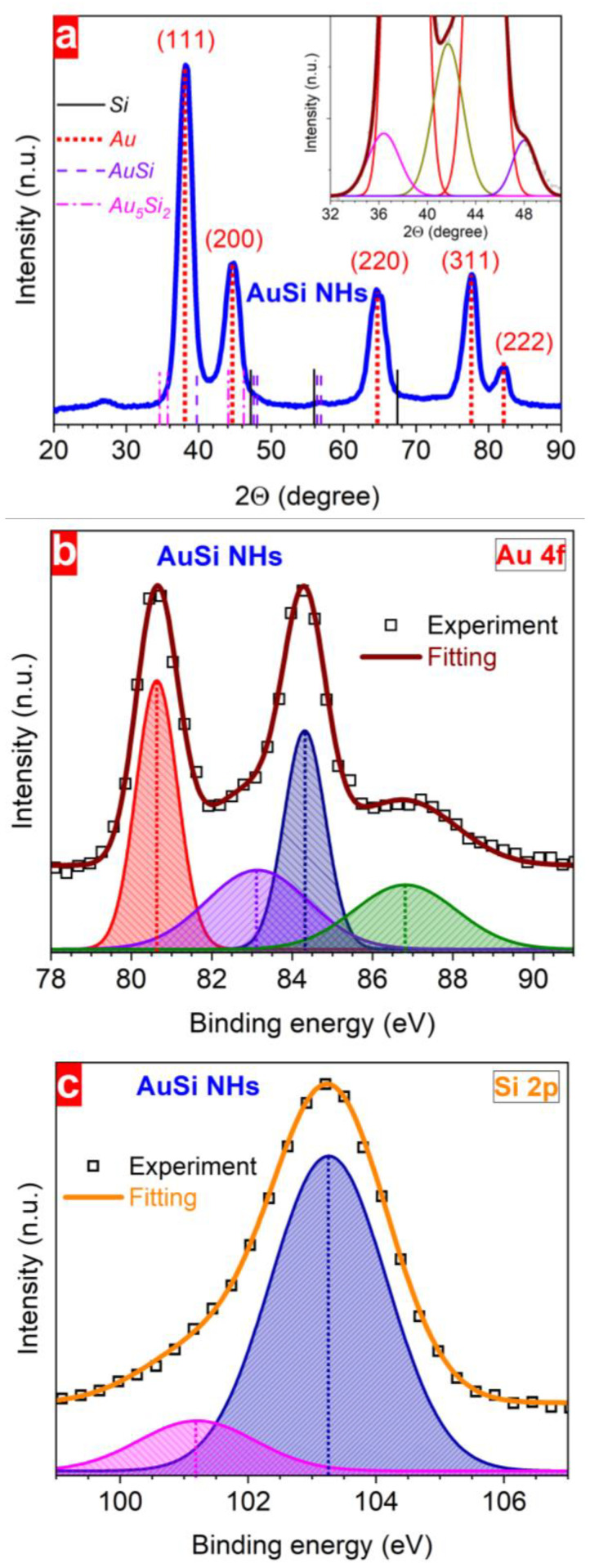
Structural properties of hybrid AuSi NPs: XRD pattern (**a**), Au 4f XPS (**b**) and Si 2p XPS (**c**) spectra. Laser fluence is 100 μJ/pulse, initial concentration of Si NPs is 30 mg/L. Vertical lines in (**a**) indicate 2Θ angles for different elemental units in AuSi NHs, demonstrating the presence of crystalline phase of gold. Inset shows zoom of the 2Θ region between 32° and 51°.

**Figure 7 nanomaterials-13-00764-f007:**
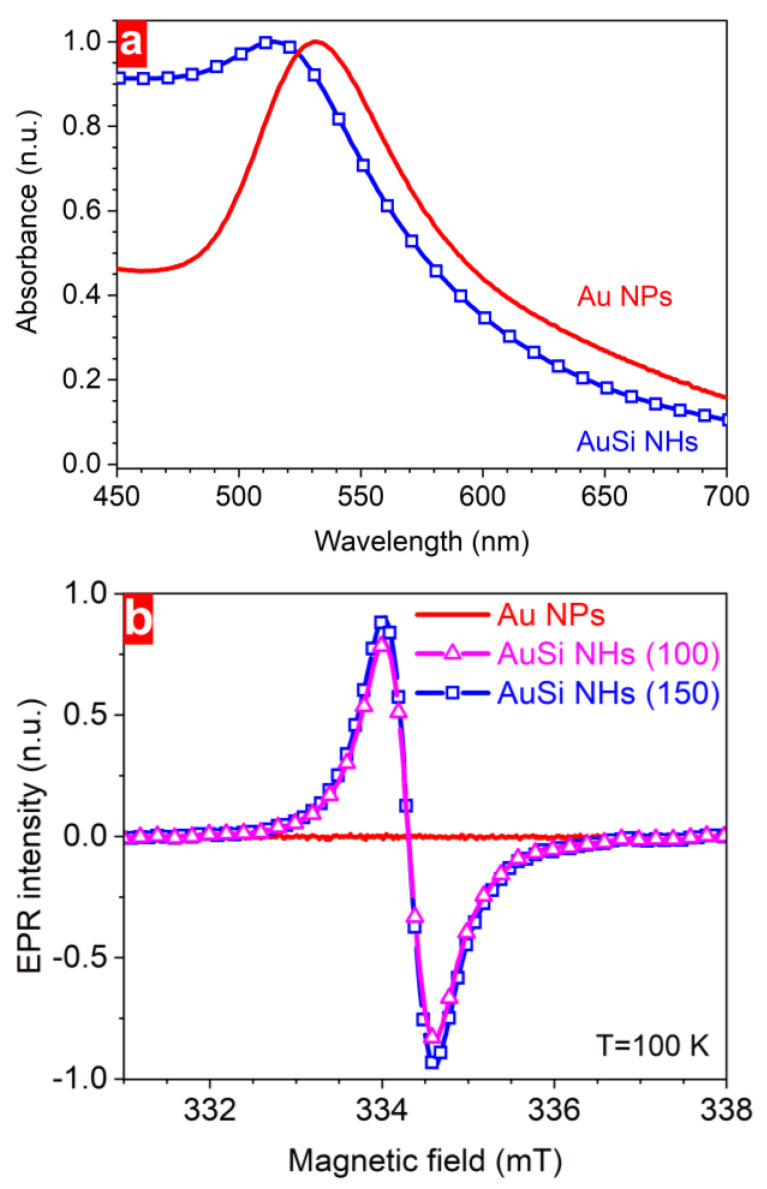
Optoelectronic properties of Au-based NPs: absorbance (**a**) and EPR (**b**) spectra.

**Figure 8 nanomaterials-13-00764-f008:**
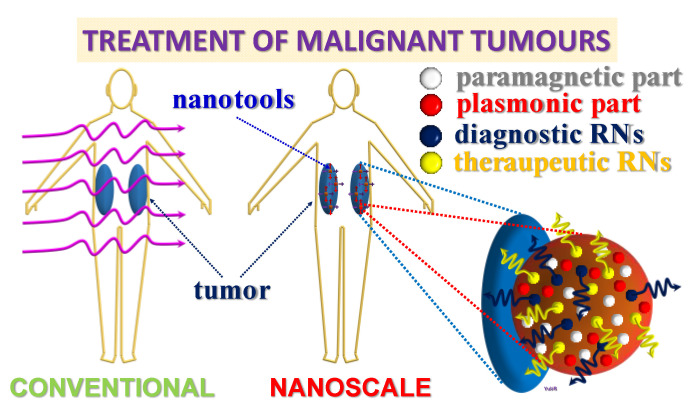
Further outlook of the employment of laser-synthesized multicomponent nanotools for biomedical applications.

## Data Availability

Not applicable.

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
