# Peer review of "Environmentally Friendly Improvement of Plasmonic Nanostructure Functionality towards Magnetic Resonance Applications"

_nanomaterials, 2023, doi:10.3390/nano13040764_

Round 1

Reviewer 1 Report

In this manuscript, the author developed a strategy of the “green” laser-based technique to prepare the nanohybrids possessing a multi-crystalline gold structure with silicon impurities. Moreover, prepared nanohybrids of the multi-crystalline structure exhibit plasmonic and paramagnetic modalities simultaneously. This strategy opens new avenues for plasmonic nanostructures as contrast agents in optical and magnetic resonance imaging techniques, biosensing and cancer theranostics. However, there are several deficiencies in this work. After careful consideration, I think, it is possible to be accepted after a minor revision. The corresponding information or explanations should be given before possible acceptance.

1.        The authors claimed that presenced Au–Si bonds formed by a laser-induced interaction between gold and silicon, however it is vague about the description in this paper. Please confirm it in XPS, UV-Vis or other tests.

2.        The authors systematically analyzed the conditions affecting the synthesis of gold-silicon nanohybrids and showed that the ratio of numbers of gold and silicon atoms can be controlled in Fig.5. But the influence of different gold and silicon contents on the structure and morphology of the nanohybrids is not clear. If possible, please explain further the structure-activity relationship between excess gold nanoparticles and excess silicon nanoparticles.

3.        It's a striking strategy that a successful combination of metallic and semiconductor elements in one nanoparticle was achieved by ultrashort pulsed laser ablation in liquids of bulk gold in water containing silicon nanoparticles. Can silicon nanoparticles be replaced by other nanoparticles such as graphene, carbon nanotubes, black phosphorus and hybrid nanoparticles with silicon? Would the different mechanisms be triggered with different nanoparticles?

Reviewer 2 Report

The results of the synthesis and characterization of composite Au-Si nanoparticles obtained by laser ablation of gold in water containing silicon nanoparticles are presented. The motive is the prospects for the use of the such particles in biomedicine. These expectations receive certain support in view of the results of characterization performed in the work. In addition, the authors draw attention to the environmental attractiveness of the method they used. The material can be accepted, but before the authors should take into account some comments.

1. It looks unusual that the mass content of silicon in the particles is quite high, but it is practically not detected in the XRD pattern (the contributions of the Au-Si compounds also make a small contribution to the XRD pattern). The authors attribute this to the amorphous structure of silicon. One would expect some indication of this in the form of a halo (very broad peak) in the XRD pattern, but this is not observed. Could there be other explanations for this? For example, can gold particles shield X-ray, etc. "hide" silicon inclusions?

2. The terms "multicrystalline structure", "multicrystalline nature" are unfortunate. In accordance with the context, it is better to replace them with convenient "polycrystalline" (if you are talking about the fact that a gold particle contains several crystallites) or "multiphase" (if there are several phases).

3. Silicon can be considered as a potentially undesirable component for theranostics due to the risks associated with its appearance in the body (eg Silicosis). If the authors managed to obtain particles where silicon is hidden inside gold particles, this may look like an additional advantage of the presented product.
